# The Role of Z-disc Proteins in Myopathy and Cardiomyopathy

**DOI:** 10.3390/ijms22063058

**Published:** 2021-03-17

**Authors:** Kirsty Wadmore, Amar J. Azad, Katja Gehmlich

**Affiliations:** 1Institute of Cardiovascular Sciences, College of Medical and Dental Sciences, University of Birmingham, Birmingham B15 2TT, UK; KXW047@student.bham.ac.uk (K.W.); A.J.S.Azad@bham.ac.uk (A.J.A.); 2Division of Cardiovascular Medicine, Radcliffe Department of Medicine and British Heart Foundation Centre of Research Excellence Oxford, University of Oxford, Oxford OX3 9DU, UK

**Keywords:** α-actinin 2, filamin C, myopalladin, myotilin, telethonin, Z-disc alternatively spliced PDZ-motif (ZASP), myopathy, cardiomyopathy, missense variant, truncating variant

## Abstract

The Z-disc acts as a protein-rich structure to tether thin filament in the contractile units, the sarcomeres, of striated muscle cells. Proteins found in the Z-disc are integral for maintaining the architecture of the sarcomere. They also enable it to function as a (bio-mechanical) signalling hub. Numerous proteins interact in the Z-disc to facilitate force transduction and intracellular signalling in both cardiac and skeletal muscle. This review will focus on six key Z-disc proteins: α-actinin 2, filamin C, myopalladin, myotilin, telethonin and Z-disc alternatively spliced PDZ-motif (ZASP), which have all been linked to myopathies and cardiomyopathies. We will summarise pathogenic variants identified in the six genes coding for these proteins and look at their involvement in myopathy and cardiomyopathy. Listing the Minor Allele Frequency (MAF) of these variants in the Genome Aggregation Database (GnomAD) version 3.1 will help to critically re-evaluate pathogenicity based on variant frequency in normal population cohorts.

## 1. Introduction

Myopathies (myo—muscle, pathy—suffering) are diseases of the skeletal muscle, which are frequently hereditary. Patients present with a wide variety of clinical phenotypes, which often overlap, but are usually characterised by progressive muscle weakness due to the degeneration of muscle. Myopathies can include proximal and distal myopathy; diseases affecting the proximal muscles (muscles which are close to the centre of the body—shoulder, pelvis, and upper arms) and lower limbs, respectively. Limb-girdle muscular dystrophy (LGMDA) is a proximal muscle disease affecting the proximal limb and girdle muscles (muscles around the hips and shoulders). Patients with such dystrophies present with muscle atrophy, pseudohypertrophy and contractures of the joints, which can consequently lead to impaired or even total loss of movement. Loss of ambulation means patients may become wheelchair-bound. If the diaphragm is affected, lung function might be decreased and their oxygen intake reduced, and they may need to be ventilated. At the histopathological level, myofibrillar myopathies (MFM), which are among the most common skeletal muscle myopathies, commonly display protein aggregation due to a significant level of myofibrillar disorganisation. Characterisation of MFM protein aggregates highlighted the presence of Z-disc proteins such as filamin C, myotilin an Z-disc alternative spliced PDZ-motif (ZASP) just to name a few (reviewed in [1]). Other forms are nemaline myopathy (NM), which is characterised by the presence of ‘nemaline bodies’ and rod-like structures. Though generally classified as diseases of the skeletal muscle, patients with myopathies may additionally present with a cardiac phenotype (‘cardiac involvement’) [2].

Diseases that primarily affect the cardiac muscle are termed cardiomyopathies. They are heterogeneous myocardial disorders affecting the structure and function of the heart muscle in the absence of coronary artery disease, hypertension, and congenital heart disease [3]. Cardiomyopathies are classified into the following categories: dilated (DCM), hypertrophic (HCM), restrictive (RCM), arrhythmogenic (ACM) cardiomyopathies and left ventricular noncompaction (LVNC). DCM is primarily characterised by a dilation of the left ventricle with poor contraction force (systolic dysfunction). HCM is characterised by the enlargement of cardiac mass (hypertrophy), which is accompanied by impaired relaxation (diastolic dysfunction). HCM is the most common cardiomyopathy with a prevalence of 1:500 [4]. RCM is characterised by a stiffening of the inner ventricular walls, also leading to diastolic dysfunction. ACM is an arrhythmic cardiomyopathy with right, left or biventricular impairment of systolic function. LVNC is characterised by pronounced left ventricle trabeculations. Some cardiomyopathies are associated with a high risk of sudden cardiac death due to ventricular arrhythmias (especially HCM and ACM).

There is phenotypic overlap between these diseases, e.g., ACM can affect primarily the left ventricle, mimicking DCM with arrhythmias [5]. Likewise, both RCM and HCM are characterised by diastolic dysfunction [6]. For a more detailed description of clinical and molecular features of myopathies and cardiomyopathies see excellent reviews [7,8].

The striated muscle cells, myotubes in skeletal muscle and cardiomyocytes in the heart share the general architecture of their contractile apparatus: organised in myofibrils, the smallest contractile units are the sarcomeres, in which actin-based thin filaments and myosin-based thick filaments are assembled. The sliding of these filaments against each other generates force and results in the contraction of the muscle [9]. Thick filaments are anchored by a structure called the M-band [10], while thin filament are anchored by the Z-disc [11]. Titin, the third filament, connects Z-discs and M-bands by spanning half a sarcomere [12].

The roles of titin and M-band proteins in striated muscle diseases have been highlighted in excellent reviews, e.g., [10,13]. This review will focus on Z-disc proteins and their role in striated muscle diseases. We have selected Z-disc proteins that are implicated in genetic disease of both skeletal muscle and the heart, or cause myopathy with cardiac involvement. The following text will therefore focus on α-actinin, filamin C, myopalladin, myotilin, telethonin and ZASP.

## 2. Z-disc Proteins in Myopathy and Cardiomyopathy

### 2.1. α-actinin 2

α-actinin is a protein found in the Z-disc of muscles and is involved in the cross-linking of anti-parallel actin filaments. It also interacts with titin and acts as a protein scaffold for large protein complexes [14]. The protein has four different isogenes (*ACTN1-4*), that are all closely related and perform similar functions, but display different expression profiles across tissues. The binding of non-muscle α-actinin (*ACTN1* and *ACTN4*) to actin is regulated by the binding of calcium, whilst in muscle isoforms (*ACTN2* and *ACTN3*) binding to actin is calcium insensitive [15]. Instead, muscle α-actinin is regulated by the binding of phospholipids such as phosphatidylinositol bisphosphate [16]. *ACTN2* (located at chromosome position 1q42-43) is highly expressed in heart and skeletal muscle, while *ACTN3* (located at chromosome position 11q13.2) is exclusively expressed in skeletal muscle.

α-actinin forms antiparallel homodimers [16]. It is comprised of an actin binding domain (ABD), a central rod domain with 4 spectrin repeats, and an EF hand domain (Figure 1). Through its binding to a variety of other signalling molecules, it has an important role in organisation of the cytoskeleton and muscle contraction [14].

A loss of function study of *ACTN2* in zebrafish embryos using antisense morpholino technology highlighted the importance of α-actinin 2 in Z-disc lateral alignment in zebrafish cardiogenesis [17]. Depletion of *ACTN2* led to a reduced end-diastolic diameter and overall reduced cardiac function. The loss of function mutants also had reduced cardiomyocyte size and number, leading to reduced ventricular chamber size. This underlines the crucial role of α-actinin for integrity in the developing heart.

#### 2.1.1. α-actinin 2 in Cardiomyopathy

Several studies report links between variants (*ACTN2*) and HCM. The first description was by Theis et al. [18]. As part of a genetics candidate gene approach, they interrogated 5 genes coding for Z-disc-proteins, including *ACTN2*, in 239 unrelated patients with HCM by polymerase chain reaction (PCR), denaturing high performance liquid chromatography (DHPLC) and direct Sanger DNA sequencing. They identified two pathogenic variants in *ACTN2* (p.Gly111Val and p.Arg759Thr), and also reported *ACTN2* p.Thr495Met. This variant (*ACTN2* p.Thr495Met), was initially thought to be pathogenic, but now appears to be relatively common [19] in normal populations to support a pathogenic role (Table 1). Echocardiography suggested that pathogenic variants in *ACTN2* (as well as other HCM-associated Z-disc genes) resulted in sigmoid septal morphology of cardiac hypertrophy. 

A separate study looked at 23 individuals from a 3-generation Australian family that clinically presented with heterogeneous HCM [20]. A genome-wide linkage analysis, combined with candidate gene screening, identified *ACTN2* p. Ala119Thr. The variant was found in all 7 affected members of the HCM family. To further support the link between HCM and pathogenic variants in *ACTN2*, 297 additional probands were screened for variants in *ACTN2*. High-resolution melt analysis of four HCM families identified an additional two *ACTN2* variants (p.Glu583Ala, and p.Glu628Gly).

The *ACTN2* variant p.Met228Thr was identified in an Italian family with atypical HCM and early onset atrial fibrillation (AF) by next generation sequencing [21]. It co-segregated with disease in 11 affected family members. *ACTN2* p.Thr247Met, another variant located in the actin-binding domain, was identified in a patient who presented with familial HCM [22]. A human-induced pluripotent stem cell (hiPSC)-derived cardiomyocyte model for the variant supported its pathogenicity and suggested that the observed prolongation of QT interval in the family could be explained by the reduced interaction of the *ACTN2* p.Thr247Met variant with the L-type calcium channel. This observation lead to the successful treatment of affected family members with L-type calcium channel blocker diltiazem. 

In summary, most *ACTN2* identified in HCM patients are located in the actin-binding domain (see Table 1). The crystal structures of two HCM-associated *ACTN2* variants (p.Ala119Thr and p.Gly111Val) located in the actin-binding domain, showed small, yet distinctive changes [23]. The *ACTN2* p.Gly111Val variant reduced thermal stability and impacted tertiary structure of the protein, whilst the *ACTN2* p.Ala119Thr had a more pronounced negative effect on actin binding. The variants showed weaker integration into the Z-discs than wild-type α-actinin and formed aggregates outside the Z-discs. They further showed reduced ability to leave the Z-disc with reduced fluorescent decay after photoactivation, suggesting that exchange of the protein at Z-discs is reduced in the presence of the pathogenic variants.

Associations of *ACTN2* variants with other types of cardiomyopathies are less strong: *ACTN2* p.Gln9Arg variant was identified in the DCM patient [24]. Despite demonstrating altered binding properties to muscle LIM Protein in vitro and being absent in 200 control individuals, the variant would not pass current criteria of pathogenicity (based on its frequency in normal cohorts, Table 1). 

More recently, *ACTN2* p.Leu320Arg was identified by whole-exome sequencing in a Chinese family, who presented with DCM and ventricular tachycardia [25]. 12 family members were followed up and four of them were affected by disease. All four carried had the *ACTN2* p.Leu320Arg variant that co-segregated with disease and was absent in the healthy family members. 

A further study associated the *ACTN2* p.Ala119Thr with disease in an Australian family who showed cardiac phenotypic heterogeneity (LVNC or DCM) [26]. Using exome sequencing in two probands, the variant *ACTN2* p.Ala119Thr was ranked the most highly expressed in cardiac tissue according to data derived from RNAseq and found to co-segregate with disease of the four affected family members. Of note, the same change had previously been identified in HCM patients (see above) and it is not understood why it can cause different types of cardiomyopathy in different individuals. 

**Table 1 ijms-22-03058-t001:** Variants in the *ACTN2* gene that have been previously reported in individuals with myopathy or cardiomyopathy.

Name Gene	Variant	Het or Homo	Type of Disease	Cardiac or Muscular	* MAF on GnomAD	Location	Ref
α-actinin *ACTN2*	p.Gln9Arg	Het	DCM	Cardiac	7.22 × 10^−4^	ABD	[24]
	**p.Gly111Val**	**Het**	**HCM**	**Cardiac**	**Absent**	**ABD**	[18]
	**p.Ala119Thr**	**Het**	**HCM, LVNC, DCM**	**Cardiac**	**Absent**	**ABD**	[20,26]
	**p.Leu131Pro**	**Het**	**DM**	**Muscular**	**Absent**	**ABD**	[27]
	**p.Met228Thr**	**Het**	**HCM (with AF)**	**Cardiac**	**Absent**	**ABD**	[21]
	**p.Thr247Met**	**Het**	**HCM**	**Cardiac**	**Absent**	**ABD**	[22]
	**p.Leu320Arg**	**Het**	**DCM (with VT)**	**Cardiac**	**7.95 × 10^−6^**	**SR1**	[25]
	**p.Cys487Arg**	**Het**	**DM**	**Muscular**	**Absent**	**SR2**	[27]
	p.Thr495Met	Het	HCM	Cardiac	2.76 × 10^−4^	SR2	[18]
	**p.Glu583Ala**	**Het**	**HCM**	**Cardiac**	**Absent**	**SR3**	[20]
	**p.Glu628Gly**	**Het**	**HCM**	**Cardiac**	**Absent**	**SR3**	[20]
	**p.Arg759Thr**	**Het**	**HCM**	**Cardiac**	**Absent**	**EF1**	[18]

Abbreviations: ABD, actin-binding domain; AF, atrial fibrillation; DCM, dilated cardiomyopathy; DM, distal myopathy; EF, EF hand; HCM, hypertrophic cardiomyopathy; Het, heterozygous; Homo, homozygous; LVNC, left ventricular noncompaction; SR, spectrin like repeat; VT ventricular tachychardia. * MAF on GnomAD: The Minor Allele Frequency (MAF) was found on The Genome Aggregation Database (gnomAD) version 3.1. All variants with MAF < 10^−4^ are highlighted in bold.

#### 2.1.2. α-actinin in Myopathy

Thus far, only one report performing high-throughput sequencing has linked missense variants in *ACTN2* with autosomal dominant distal myopathy [27]. *ACTN2* p.Cys487Arg was identified in three families and *ACTN2* p.Leu131Pro in a further one. The missense variants in *ACTN2* co-segregated with the disease in affected family members. Clinically, they showed adult-onset asymmetric distal muscle weakness, e.g., atrophy of the Tibialis anterior muscle, which later progressed to proximal limb muscles. 

*ACTN3* p.Arg577X is a common nonsense variant (minor allele frequency, MAF, ~20%) previously described to cause α-actinin-3 deficiency when individuals are homozygous for the X allele [28]. Hogarth et al. found this genotype to be a modifier of the clinical phenotype of Duchenne muscular dystrophy patients [29]. A double knockout mouse model, lacking both dystrophin and α-actinin 3, showed reduced muscular strength but was protected from stretch-induced damage with age by an increase in oxidative muscle metabolism.

#### 2.1.3. Summary

Pathogenic variants in *ACTN2* are rare and do not significantly contribute to cardiomyopathies in numbers. *ACTN2*’s association with HCM is best documented. Until now, there is only one report linking *ACTN2* to myopathy. In light of α-actinin2’s conservation and documented requirement for cardiac integrity [17], it is likely that most missense variants in *ACTN2* are detrimental to its crucial functions and hence not compatible with life. Nevertheless, the few identified ones are of mechanistic interest.

### 2.2. Filamin C

Filamin C is a member of the actin-binding filamin family, consisting of three members: filamin A, B and C. Filamins bind to more than 90 different binding partners and are thought to provide structural stability and to mediate crucial signalling functions. Filamin C is found in the Z-discs and costameres of myotubes and cardiomyocytes and at the intercalated discs of cardiomyocytes, i.e., at load-bearing sites of the cardiac and skeletal muscle cells. It is proposed to play a role in mechanosensing [30,31,32].

Encoded on chromosome 7q32.1, filamin C consists of an amino-terminal actin binding domain followed by 24 immunoglobulin-like (Ig) domains (Figure 2) [33,34], which are structured into two rod region (Ig1-15 Rod1, Ig16-23 Rod2) [33]. These flexible rod regions regulate the spacing and orientation of F-actin cross-linkages through protein-protein interactions with cytoskeletal, transmembrane and cell signalling proteins [35,36].

Ig20 contains an 82 amino acid insertion unique only to filamin C. It mediates muscle-specific ligand interactions with, e.g., myotilin [37], and is thought to assist in myofibrillogenesis, and presumably plays a role in muscle maintenance [38,39,40]. These ligand interactions include, but are not necessarily limited to roles such as autophagy control and Z-disc organisation [41,42]. The carboxy-terminal portion of the protein consists of the dimerization domain Ig24, which binds to membrane glycoproteins and allows homodimerization of *FLNC* to form and maintain a network of cross-linking F-actin filaments that link membrane proteins to actin [34,43]. In addition to modulating actin dynamics within the Z-disc, ligand interactions with HSPB7 and calpain 1 around Ig 24 have also been identified [44,45].

In addition, filamin C plays a role in maintaining structural stability of the costameric sarcolemma via interactions with sarcoglycans on the dystrophin-glycoprotein complex [46]. Signalling roles of filamin C includes regulating the morphology and migration of myogenic progenitor cells via binding to cell-matrix adhesion protein migfilin [47]. These processes of the protein can be modulated by phosphorylation events [48,49].

Skeletal muscle functions of filamin C are well established, highlighting its role in chaperone-assisted selective autophagy. However, filamin C’s functions within the cardiac muscle are relatively unknown. In part this could be due to early stage lethality in *FLNC* knockout models potentially masking any cardiac implications [50,51]. This has been overcome by an inducible cardiomyocyte-specific *FLNC* knockout mouse model, which displays high rates of lethality along with cardiac defects, including heart dilation, extensive fibrosis and systolic dysfunction [52].

#### 2.2.1. Filamin C in Cardiomyopathy

Due to much higher levels of *FLNC* expression in skeletal muscle compared to cardiac muscle (3.5-fold) [53], variants found in this protein have predominantly been associated with neuromuscular conditions such as MFM [43,54,55,56] (see Section 2.2.2). However, in these cases a cardiac role cannot be excluded since most studies on skeletal muscle-related filaminopathies did not thoroughly characterise cardiac phenotypes (though it is rare for cardiomyopathy patients to have a skeletal phenotype with *FLNC* variants) [54,57,58].

Independent of skeletal muscle involvement, pathogenic *FLNC* variants have been identified in familial HCM, RCM and ACM cases [59,60]. They have also been identified by sequencing in familial DCM, ACM, LVNC, cardiac arrhythmias and arrhythmogenic bileaflet mitral valve prolapse syndrome, with now 101 variants being reported (Table 2) [56,61,62,63]. Patients carrying *FLNC* variants have a higher risk of sudden cardiac death compared to patients without *FLNC* variants [59,60].

It has been suggested that the subtype of cardiomyopathy is determined by the variant type and site location (Table 2) as seen for other cardiomyopathy-associated genes (e.g., *DES*, *LDB3* [64], *BAG3* [65], *FHL1* [66]) [67]. 95% of *FLNC* variants found in HCM patients are missense variants whilst frameshift, in-frame, nonsense and splice variants resulting in a loss of function contribute to 81% of DCM-related filaminopathies [68]. Two missense variants have been described in DCM, although one was a case of neonatal DCM resulting in a null variant [69].

In the healthy heart, filamin C localises at the Z-discs of the sarcomeres and at the intercalated discs, where it links actin structures with the cytoskeleton. However, immunohistochemical assessments of cardiomyopathy patients with *FLNC* missense variants reported the loss of filamin C from the intercalated disc [60,67,69]. There is mixed evidence about the role of filamin C aggregation in cardiomyopathy [61,62]. The amino acid change and location of a *FLNC* variant may render it susceptible to unfolding and aggregation or not [59,60,61,62,69,70,71,72]. Whilst filamin C aggregates have not been observed in DCM patients [61], they have been associated with *FLNC* missense variants (*FLNC* p.Ser1624Leu, p.Ile2160Phe, p.Val2297Met) in patients with HCM, cardiac arrhythmias and RCM [57,58,60,70,73].

Overexpression of a missense *FLNC* variant in the zebrafish model resulted in myofibrillar protein aggregation and of irregular *FLNC* distribution along the Z-disc [67]. Cellular models (rat cardiac myoblasts, H9C2, and mouse C2C12 myoblasts) also exhibited protein aggregation in the presence of HCM-related missense variants, (*FLNC* p.Val123Ala, p.Ala1539Thr, p.Arg2133His and p.Ala2430Val) [59]. In some missense variants ((p.Ala1539Thr and p.Arg2133His), actin aggregates were present rather than filamin C aggregates suggesting that these filamin C variants alter folding and result in cytoskeleton rearrangements [59].

The pathogenic RCM-associated variant *FLNC* p.Val2297Met, located in Ig20, was investigated in a stem cell-derived cardiomyocyte model [70]. This variant exhibited reduced fractional shortening and patient-observed relocalisation to suggest it acts as a loss-of-function variant [70].

#### 2.2.2. Filamin C in Myopathy

In agreement with the severe skeletal muscle defects observed when *FLNC* is inactivated in mice, *FLNC* has been implicated in human skeletal muscle pathologies [50]. These skeletal myopathies have features of progressive degeneration of the muscle fibres and are characterised by protein aggregation and the presence of rimmed vacuoles in muscle fibres.

In 2005, Vorgerd et al. [54] discovered the first pathogenic *FLNC* variant, *FLNC* p.Trp2710X, in patients with MFM characteristics [54]. This nonsense *FLNC* variant interferes with dimerisation of Ig24, due to the deletion the carboxy-terminal 16 amino acids [74] and has been identified as an ethnically independent variant hotspot [75]. Almost one-third of patients carrying the *FLNC* p.Trp2710X variant also presented with cardiac abnormalities. Patients carrying this variant or the later described deletion variants in Ig7 (*FLNC* p.Val930_Thr933del, p.Lys899_Val904del/Val899_Cys900ins, p.931_935del [76,77]) exhibited defects in muscle structure and redistribution of filamin C into protein aggregates and Z-disc streaming in animal models [74]. However, in some distal filaminopathy and MFM cases, protein aggregates are absent despite evidence of fiber disintegration leading to muscle weakness. Whether protein aggregation occurs (as in *FLNC* p.Trp2710X [78]) or there is disruption to *FLNC*’s interactions with its binding partners (as in *FLNC* p.Ala193Thr and p.Met251Thr [55]), there is a resultant deficiency of *FLNC* at the Z-disc, which results in Z-disc destabilisation and the consequent fiber disintegration observed in MFM and DM.

Filamin C is a client target of the chaperone-assisted selective autophagy system [79], which identifies and clears misfolded filamin C from the cells and initiates synthesis of new filamin C molecules. It is thought that the variants induce aggregation by disrupting this autophagy clearance mechanism [78,80].

*FLNC* missense and frameshift variants have also been shown to cause distal myopathy—a disease characterised by muscle weakness in the limbs [81]. MFM with lower limb involvement characterised by myofibrillar aggregation was reported in *FLNC* p.Val2375Ile and p.Met222Val [75]. However, the distal and proximal myopathy-related missense variants in the amino-terminal actin-binding domain of filamin C (*FLNC* p.Ala193Thr, p.Met251Thr) increased filamin C’s affinity to F-actin, resulting in F-actin aggregates [55]. As found in the *FLNC* p.Ala193Thr, p.Arg159His and p.Val831Ile variants, filamin C expression is closely linked to neurodegeneration and aberrant filamin C expression is present in neuronal disease such as frontotemporal dementia [56,82].

Guergueltcheva et al. demonstrated reduced muscular *FLNC* transcript levels in the presence of *FLNC* p.Phe1720LeufsX63 via nonsense-mediated mRNA decay [56]. They concluded that *FLNC* haploinsufficiency can result in a specific type of myopathy—distal myopathy with upper limb predominance [56].

#### 2.2.3. Summary

Variants found in *FLNC* lead to both skeletal and cardiac myopathies. The genetic location and type of variant can explain the different phenotypes patients present with (Table 2). For instance, *FLNC* variants in the dimerisation region of filamin C leads to the formation of aggregates found in MFM. Although clearly involved in skeletal myopathies, the overlapping phenotypes observed in *FLNC* variants affecting the actin-binding domain clouds the elucidation of any genotype-phenotype relationship [81]. Moreover, it is not current clear why some *FLNC* truncations can cause skeletal muscle defects yet others result in exclusive cardiac phenotypes.

**Table 2 ijms-22-03058-t002:** Variants in the *FLNC* gene that have been previously reported in individuals with myopathy or cardiomyopathy.

Name Gene	Variant	Het or Homo	Type of Disease	Cardiac or Muscular	* MAF on GnomAD	Location	Ref
Filamin C *FLNC*	**p.Tyr7ThrfsX51**	**Het**	**DCM**	**Cardiac**	**Absent**		[83]
	**p.Trp34X**	**Het**	**ABiMVP**	**Cardiac**	**Absent**		[84]
	**p.Gln39X**	**Het**	**DCM**	**Cardiac**	**Absent**		[71]
	**p.Gly54Asp**	**Het**	**PM**	**Muscular**	**Absent**	**ABD/CH1**	[85]
	**p.Leu77ProfsX73**	**Het**	**DCM**	**Cardiac**	**Absent**		[71]
	**p.Arg81AlafsX15**	**Het**	**DCM**	**Cardiac**	**Absent**		[61]
	**p.Tyr83X**	**Het**	**DCM**	**Cardiac**	**Absent**		[61]
	**p.Phe106Leu**	**Het**	**DCM**	**Cardiac**	**3.94 × 10^−5^**	**ABD/CH1**	[69]
	**p.Glu108X**	**Het**	**HCM**	**Cardiac**	**Absent**		[59]
	**p.Val123Ala**	**Het**	**HCM**	**Cardiac**	**Absent**	**ABD**	[59]
	**p.Val123Met**	**Het**	**DCM**	**Cardiac**	**6.57 × 10^−6^**	**ABD**	[68]
	**p.Ala193Thr**	**Het**	**DM, PM, CNS**	**Muscular**	**6.57 × 10^−6^**	**ABD/CH1**	[55]
	**p.Leu194ProfsX52**	**Het**	**DCM**	**Cardiac**	**Absent**		[61]
	**p.Gly201ValfsX36**	**Het**	**DCM**	**Cardiac**	**Absent**		[61]
	**p.Met222Val**	**Het**	**DM, PM, MFM**	**Muscular**	**Absent**	**ABD/CH1**	[86]
	**p.Gln233X**	**Het**	**DCM**	**Cardiac**	**Absent**		[68]
	**p.Glu238ArgfsX14**	**Het**	**DCM**	**Cardiac**	**Absent**		[83]
	**p.Met251Thr**	**Het**	**DM, PM, CNS**	**Muscular**	**Absent**	**ABD/CH1**	[55]
	**p.Arg269X**	**Het**	**DCM**	**Cardiac**	**Absent**		[87]
	**p.Glu309Lys**	**Het**	**IBM**	**Muscular**	**3.94 × 10^−5^**	**ABD/CH1**	[88]
	**p.Pro442Arg**	**Homo**	**Muscle weakness**	**Muscular**	**Absent**	**ROD1/Ig2**	[89]
	**p.Arg482X**	**Het**	**DCM**	**Cardiac**	**Absent**		[90]
	**p.Val489GlyfsX33**	**Het**	**DCM**	**Cardiac**	**Absent**		[68]
	p.Arg526Gln	Het	IBM	Muscular	1.58 × 10^−3^	ROD1/Ig3	[88]
	**p.Gln572X**	**Het**	**DCM**	**Cardiac**	**Absent**		[61]
	**p.Arg575Trp**	**Het**	**IBM**	**Muscular**	**Absent**	**ROD1/Ig4**	[88]
	**p.Tyr630X**	**Het**	**DCM**	**Cardiac**	**Absent**		[71]
	**p.Asp646Asn**	**Het**	**OM**	**Muscular**	**3.94 × 10^−5^**	**ROD1/Ig4**	[68]
	**p.Asp648Tyr**	**Het**	**PM, DM, MFM**	**Muscular**	**Absent**	**ROD1/Ig4**	[91]
	**p.Ala656ProfsX8**	**Het**	**ACM/SCD**	**Cardiac**	**Absent**		[63]
	**p.Ile683ArgfsX9**	**Het**	**DCM**	**Cardiac**	**Absent**		[83]
	**p.Asp693Ala**	**Het**	**IBM**	**Muscular**	**2.87 × 10^−3^**	**ROD1/Ig5**	[88]
	**p.Tyr705X**	**Het**	**ACM/SCD**	**Cardiac**	**Absent**		[63]
	**p.Gln707X**	**Het**	**DCM**	**Cardiac**	**Absent**		[87]
	**p.Lys737SerfsX11**	**Het**	**DCM**	**Cardiac**	**Absent**		[61]
	**p.Val831Ile**	**Het**	**Pick’s Disease**	**Muscular**	**2.63 × 10^−5^**	**ROD1/Ig6**	[82]
	**p.Tyr928X**	**Het**	**DCM**	**Cardiac**	**Absent**		[83]
	**p.Pro963ArgfsX26**	**Het**	**DCM**	**Cardiac**	**Absent**		[61]
	**p.Arg991X**	**Het**	**DCM**	**Cardiac**	**Absent**		[69]
	**p.Gln1024X**	**Het**	**LVNC**	**Cardiac**	**Absent**		[92]
	**p.Asp1061IlefsX17**	**Het**	**DCM**	**Cardiac**	**6.57 × 10^−6^**		[68]
	**p.Glu1104X**	**Het**	**DCM**	**Cardiac**	**Absent**		[71]
	**p.Phe1135AlafsX62**	**Het**	**DCM**	**Cardiac**	**Absent**		[61]
	**p.Ala1183Leu**	**Het**	**RCM**	**Cardiac**	**Absent**	**ROD1/Ig10**	[67]
	**p.Ala1186Val**	**Het**	**PM, DM, CM**	**Muscular**	**Absent**	**ROD1/Ig10**	[93]
	**p.Val1198GlyfsX64**	**Het**	**DCM**	**Cardiac**	**Absent**		[83]
	**p.Tyr1216Asn**	**Het**	**PM, MFM, Car**	**Muscular**	**Absent**	**ROD1/Ig10**	[73]
	**p.Tyr1230Asn**	**Het**	**Myopathy, Car**	**Muscular**	**Absent**	**ROD1/Ig10**	[94]
	**p.Pro1236Ser**	**Het**	**PM, Car**	**Muscular**	**Absent**	**ROD1/Ig10**	[95]
	p.Arg1241Cys	Het	IBM	Muscular	6.94 × 10^−3^	ROD1/Ig10	[88]
	**p.Leu1280ProfsX52**	**Het**	**DCM**	**Cardiac**	**Absent**		[68]
	**p.Arg1354X**	**Het**	**DCM**	**Cardiac**	**Absent**		[71]
	**p.Asn1369LysfsX36**	**Het**	**DCM**	**Cardiac**	**Absent**		[96]
	**p.Arg1370X**	**Het**	**ACM/SCD**	**Cardiac**	**Absent**		[63]
	**p.Gly1424Val**	**Het**	**HCM**	**Cardiac**	**6.57 × 10^−6^**	**ROD1/Ig12**	[83]
	**p.Ala1539Thr**	**Het**	**HCM**	**Cardiac**	**Absent**	**ROD1/Ig14**	[59]
	**p.Leu1573X**	**Het**	**DCM**	**Cardiac**	**Absent**		[97]
	**p.Ser1624Leu**	**Het**	**RCM**	**Cardiac**	**Absent**	**ROD1/Ig14**	[60]
	**p.Phe1626Ser fsX40**	**Het**	**DCM/SCD**	**Cardiac**	**Absent**		[98]
	**p.Asp1691Asn**	**Het**	**PM, DM, MFM**	**Muscular**	**5.25 × 10^−5^**	**ROD1/Ig15**	[91]
	p.Gly1760Ser	Het	PM	Muscular	1.12 × 10^−4^	ROD2/Ig16	[95]
	**p.Gly1800X**	**Het**	**DCM**	**Cardiac**	**Absent**		[61]
	**p.Tyr1840X**	**Het**	**DCM**	**Cardiac**	**Absent**		[99]
	**p.Gly1891ValfsX62**	**Het**	**DCM**	**Cardiac**	**Absent**		[87]
	**p.Gly2039Arg**	**Het**	**HCM**	**Cardiac**	**6.57 × 10^−6^**	**ROD2/Ig19**	[83]
	**p.Gly2070Ser**	**Het**	**DCM**	**Cardiac**	**Absent**	**ROD2/Ig19**	[61]
	**p.Ser2077ArgfsX50**	**Het**	**DCM**	**Cardiac**	**Absent**		[96]
	**p.Pro2081LeufsX2**	**Het**	**DCM**	**Cardiac**	**Absent**		[61]
	**p.Ile2086GlnfsX3**	**Het**	**DCM**	**Cardiac**	**Absent**		[68]
	**p.Arg2133Cys**	**Het**	**HCM**	**Cardiac**	**Absent**	**ROD2/Intradomain**	[59]
	**p.Arg2133His**	**Het**	**HCM**	**Cardiac**	**6.57 × 10^−6^**	**ROD2/Intradomain**	[59]
	**p.Arg2140Gln**	**Het**	**HCM**	**Cardiac**	**6.61 × 10^−6^**	**ROD2/Intradomain**	[72]
	**p.Ile2160Phe**	**Het**	**RCM**	**Cardiac**	**Absent**	**ROD2/Intradomain**	[60]
	**p.Arg2176Cys**	**Het**	**IBM**	**Muscular**	**Absent**	**ROD2/Intradomain**	[100]
	**p.Arg2187Pro**	**Het**	**PM, Car**	**Muscular**	**Absent**	**ROD2/Intradomain**	[68]
	**p.Glu2189X**	**Het**	**ARVC**	**Cardiac**	**Absent**		[101]
	**p.Thr2238Ile**	**Het**	**Myopathy, CM**	**Muscular**	**Absent**	**ROD2/Intradomain**	[68]
	p.Glu2270Lys	Het	OM	Muscular	6.40 × 10^−4^	ROD2/Ig20	[68]
	**p.Ser2275Ile**	**Het**	**OM**	**Muscular**	**Absent**	**ROD2/Ig20**	[68]
	**p.Val2290ArgfsX23**	**Het**	**DCM**	**Cardiac**	**Absent**		[68]
	**p.Val2297Met**	**Het**	**RCM**	**Cardiac**	**Absent**	**ROD2/Ig20**	[70]
	**p.Pro2298Leu**	**Het**	**RCM**	**Cardiac**	**6.57 × 10^−6^**	**ROD2/Ig20**	[102]
	**p.Pro2298Ser**	**Het**	**HCM**	**Cardiac**	**Absent**	**ROD2/Ig20**	[72]
	**p.Pro2301Ala**	**Het**	**HCM**	**Cardiac**	**Absent**	**ROD2/Ig20**	[72]
	**p.Gln2303X**	**Het**	**DCM**	**Cardiac**	**Absent**		[68]
	**p.His2315Asn**	**Het**	**HCM**	**Cardiac**	**Absent**	**ROD2/Ig21**	[59]
	**p.Arg2326X**	**Het**	**DCM**	**Cardiac**	**Absent**		[61]
	**p.Val2331ArgfsX25**	**Het**	**DCM**	**Cardiac**	**Absent**		[71]
	**p.Gly2345Glu**	**Het**	**Congenital HD**	**Cardiac**	**Absent**	**ROD2/Ig21**	[103]
	p.Arg2364His	Het	IBM	Muscular	1.65 × 10^−3^	ROD2/Ig21	[68]
	p.Arg2364His	Het	IBM	Muscular	1.65 × 10^−3^	ROD2/Ig21	[88]
	**p.Tyr2373CysfsX7**	**Het**	**DCM**	**Cardiac**	**Absent**		[83]
	**p.Val2375Leu**	**Het**	**HCM**	**Cardiac**	**Absent**	**ROD2/Ig21**	[83]
	**p.Val2375Ile**	**Het**	**PM, DM, MFM**	**Muscular**	**2.63 × 10^−5^**	**ROD2/Ig21**	[104]
	**p.Pro2393Ser**	**Het**	**LVNC**	**Cardiac**	**Absent**	**ROD2/Ig21**	[68]
	**p.Thr2419Met**	**Het**	**PM, MFM, Car, CNS**	**Muscular**	**6.57 × 10^−5^**	**ROD2/Ig22**	[105]
	**p.Ala2430Val**	**Het**	**HCM**	**Cardiac**	**9.20 × 10^−5^**	**ROD2/Ig-like 22**	[59]
	**p.Pro2470His**	**Het**	**PM,Car**	**Muscular**	**Absent**	**ROD2/Ig22**	[106]
	**p.Gln2549X**	**Het**	**DCM**	**Cardiac**	**Absent**		[83]
	**p.Cys2555X**	**Het**	**DCM**	**Cardiac**	**Absent**		[83]
	**p.Tyr2563Cys**	**Het**	**RCM**	**Cardiac**	**Absent**	**ROD2/Ig23**	[102]
	**p.Asp2703ThrfsX69**	**Het**	**DCM**	**Cardiac**	**6.58 × 10^−6^**		[61]
	**p.Trp2710X**	**Het**	**PM, DM, MFM**	**Muscular**	**Absent**	**ROD2/Ig24**	[54]

Abbreviations: ABD, actin-binding domain; ABiMVP, arrhythmogenic bileaflet mitral valve prolapse; ACM, arrhythmogenic cardiomyopathy; Car, cardiac involvement; CM, congenital myopathy; CNS, central nervous system involvement; DCM, dilated cardiomyopathy; DM, distal myopathy; HCM, hypertrophic cardiomyopathy; HD, heart disease; Het, heterozygous; Homo, homozygous; IBM, inclusion body myositis; MFM, myofibrillar myopathy; OM, other nonspecified myopathy; PM, proximal myopathy; RCM, restrictive cardiomyopathy; SCD, sudden cardiac death. * MAF on GnomAD: The Minor Allele Frequency (MAF) was found on The Genome Aggregation Database (gnomAD) version 3.1. All variants with MAF < 10^−4^ are highlighted in bold.

### 2.3. Myopalladin

Myopalladin is a protein found at the sarcomeric Z-discs and I- bands in both cardiac and skeletal muscle cells [107]. Myopalladin contains five Ig domains that are separated by six interdomain insertions (IS) (Figure 3) and IS3 contains a proline-rich region [108]. Within its carboxy-terminus, myopalladin contains binding sites for α-actinin and nebulin in skeletal muscle, and nebulette in cardiac muscle [70]. These interactions help to tether actin and titin to the Z-disc and therefore have a role in signalling when the muscle is under stress. Myopalladin also contains an amino-terminal domain that binds to the cardiac ankyrin repeat protein (CARP), which is involved in controlling gene expression in muscle [108]. *MYPN*, the gene coding for myopalladin, is located on 10q21.3.

#### 2.3.1. Myopalladin in Cardiomyopathy

A number of missense variants in *MYPN* have been reported in HCM, DCM and RCM patients in several studies [106,108,109,110,111] (Table 3). However, only two of them would pass today’s scrutiny based on frequency in normal cohorts (MAF < 10^−4^). Only these are discussed here:

*MYPN* p.Pro961Leu was identified in a DCM patient [109]. This variant is located in the third Ig domain, which mediates binding to α-actinin. Sarcomeric disorganisation was observed in an endomyocardial biopsy of the patient. *MYPN* p.Arg1088His variant was identified in a patient with familial DCM and found to co-segregate with the disease: three further family members were also heterozygous for the variant and clinically affected [111]. Immunofluorescence labelling of explanted heart tissue carrying the *MYPN* p.Arg1088His variant found that the dilated left ventricle of this patient had reduced myopalladin localisation at the Z-disc. In contrast, localisation appeared normal in the right, unaffected ventricle of the patient.

Furthermore, two truncating *MYPN* variants have been associated with cardiomyopathies. After screening and sequencing the coding region of *MYPN* in 114 patients with DCM, an insertion-deletion (*MYPN* p.Ile83fsX105) was identified in a familial case of DCM, co-segregating with disease [111]. Five family members were found to be heterozygous for the variant; two were clinically affected, two were healthy and one had an unknown clinical status.

The nonsense mutation *MYPN* p.Gln529X was identified in two siblings who presented with familial RCM [107]. It was predicted to truncate the three Ig repeats and the nebulette and α-actinin binding domains of myopalladin. Transmission electron microscopy in the siblings explanted hearts after they both received heart transplants, displayed myofibrillar disarray and degeneration. Knock-in mice carrying either a heterozygous or homozygous form of *MYPN* p.Gln526X, homologous to the human *MYPN* p.Gln529X variant, reflected aspects of RCM when investigated by echocardiography, cardiac magnetic resonance imaging and morphohistology [112]. Moreover, gene expression and molecular mechanism studies found that the truncated myopalladin protein translocated to the nucleus and persisted in the heterozygous mice. Western blot analysis showed reduced levels of Carp, Erk1/2, Smad2 and Akt but increased levels of muscle LIM Protein, desmin, periostin and osteopontin. Altered transcription of these genes appeared to facilitate fibrosis and a reduced hypertrophic response in hearts of the heterozygous mice.

#### 2.3.2. Myopalladin in Myopathy

Using a combination of homozygosity mapping and whole exome sequencing, four individuals from four separate families (from 54 screened families) were identified to have biallelic loss-of-function variants in *MYPN* [113]. All patients presented with mild, childhood-to-adult-onset NM; they displayed progressive muscle weakness and walking difficulties when they reached their 40’s. Individual 1 possessed one *MYPN* variant (p.Asn668ThrfsX25). Individual 2 had four *MYPN* variants (p.Gly1026ValfsX21, p.Gly1026AsnfsX59, p.Gly1026LeufsX57, p.Gly1026_Gln1077del) whilst the third individual had one (p.Arg377X). The fourth individual had two variants (p.Arg1057X) and (p.Arg1072X). Immunostaining and Western blot analysis of muscle samples or myotubes from the patients showed a complete absence of myopalladin in these individuals and supports the variants being loss-of-function.

In a separate study, exome sequencing identified recessive homozygous variants of *MYPN* in three individuals from two families [114]. Family 1 included two siblings who had a nonsense *MYPN* p.Arg885X variant, whilst family 2 had a *MYPN* c.3158 + 1G > A variant affecting exon 16 at the essential donor splice site. All patients presented with slow, progressive congenital cap myopathy. Western blot analysis of muscle extracts from patient 1 revealed full length myopalladin to be significantly decreased compared to control samples. Light and electron microscopy detected myopalladin and α-actinin-positive caps and rods in the muscle of the patients.

Merlini et al. investigated a family where two members had varying degrees of congenital to adult-onset muscle weakness [115]. Magnetic resonance imaging (MRI) and late gadolinium enhancement revealed cardiac involvement in one of the two siblings, which was attributed to interstitial fibrosis. Neither of them appeared to have features of cap or NM. Analysis using whole exome sequencing discovered a homologous loss-of-function single nucleotide deletion (*MYPN* p.Ser769LeufsTer92) within exon 11 of *MYPN* in the two affected family members. The unaffected parents were heterozygous for the variant whilst an unaffected sibling lacked it completely. Complete absence of full-length myopalladin and the 114kd isoform 2 was documented on Western blotting, in agreement with loss of myopalladin staining from the Z-discs of skeletal muscle on immunofluorescence. Electron microscopy showed that the patient had disrupted and fragmented Z-discs.

#### 2.3.3. Summary

*MYPN* variants have been reported in HCM, DCM, and RCM, but only two missense and two nonsense variants could be considered clearly pathogenic (Table 3). The *MYPN* p.Gln529X variant was found to cosegregate with RCM [107] and is supported by animal studies. In contrast to cardiomyopathy-associated heterozygous *MYPN* variants, homozygous or bi-allelic truncating variants in *MYPN* are linked to myopathy, particularly to cap myopathy and NM. Investigation of biopsy material suggested these are loss-of-function variants, however exact disease pathways remain to be elucidated.

**Table 3 ijms-22-03058-t003:** Variants in the *MYPN* gene that have been previously reported in individuals with myopathy or cardiomyopathy.

Name Gene	Variant	Het or Homo	Type of Disease	Cardiac or Muscular	* MAF on GnomAD	Location	Ref
Myopalladin *MYPN*	p.Tyr20Cys	Het	HCM, DCM	Cardiac	1.27 × 10^−3^	IS1	[107]
	**p.Ile83fsX105**	**Het**	**DCM**	**Cardiac**	**Absent**	**IS1**	[111]
	**p.Arg377X**	**Homo**	**NM**	**Muscular**	**Absent**		[113]
	**p.Gln529X**	**Het**	**RCM**	**Cardiac**	**4.00 × 10^−6^**	**Ig2**	[107]
	**p.Asn668ThrfsX25**	**Homo**	**NM**	**Muscular**	**Absent**		[113]
	**p.Ser769LeufsTer92**	**Homo**	**Congenital to adult-onset myopathy**	**Muscular**	**Absent**		[115]
	**p.Arg885X**	**Homo**	**Congenital cap myopathy**	**Muscular**	**6.58 × 10^−6^**		[114]
	p.Arg955Trp	Het	DCM	Cardiac	4.71 × 10^−4^	Ig3	[109]
	**p.Pro961Leu**	**Het**	**DCM**	**Cardiac**	**Absent**	**Ig3**	[109]
	**p.Gly1026ValfsX21**	**Homo**	**NM**	**Muscular**	**Absent**		[113]
	**p.Gly1026AsnfsX59**	**Homo**	**NM**	**Muscular**	**Absent**		[113]
	**p.Gly1026LeufsX57**	**Homo**	**NM**	**Muscular**	**Absent**		[113]
	**p.Gly1026_Gln1077del**	**Homo**	**NM**	**Muscular**	**Absent**		[113]
	**p.Arg1057X**	**Het**	**NM**	**Muscular**	**Absent**		[113]
	**p.Arg1072X**	**Het**	**NM**	**Muscular**	**Absent**		[113]
	**p.Arg1088His**	**Het**	**DCM**	**Cardiac**	**3.98 × 10^−6^**	**Ig4**	[111]
	p.Pro1112Leu	Het	HCM, DCM	Cardiac	3.06 × 10^−3^	Ig4	[111,116]
	p.Val1195Met	Het	DCM	Cardiac	3.05 × 10^−4^	Ig5	[111]
	**c.3158 + 1G > A**	**Homo**	**Congenital cap myopathy**	**Muscular**	**3.98 × 10^−6^**		[114]

Abbreviations: DCM, dilated cardiomyopathy; HCM, hypertrophic cardiomyopathy; Het, heterozygous; Homo, homozygous; NM, nemaline myopathy; RCM, restrictive cardiomyopathy. * MAF on GnomAD: The Minor Allele Frequency (MAF) was found on The Genome Aggregation Database (gnomAD) version 3.1. All variants with MAF < 10^−4^ are highlighted in bold.

### 2.4. Myotilin

Myotilin is a Z-disc localised thin-filament associated protein found in skeletal muscle, and to a lesser extent in cardiac tissue [117]. Along with palladin and myopalladin, myotilin belongs to a family of the Ig domain-containing actin-binding proteins. This family of sarcomeric proteins are instrumental in sarcomere assembly and maintaining regular muscle structure by binding to α-actinin and filamins to cross-link actin [108,118].

The 57 kDa protein is composed of 10 exons encoding 498 amino acids and is mapped to chromosome 5q31 [118,119]. The amino-terminus contains a serine-rich region and a 23 amino acid hydrophobic stretch region, while the carboxy-terminus contains a short non-structured tail [118]. The hydrophobic stretch region mediates the localisation of myotilin to the sarcolemmal membrane (Figure 4) [120]. Myotilin is also composed of two Ig domains, which are required for homodimerisation [118]. Myotilin engages in molecular interactions with actin monomers and Z-disc proteins through its Ig domains [121], it binds to α-actinin, filamins, FATZ/myozenin/calsarcin and Engima proteins (e.g., ZASP) [118,121,122,123,124]. Myotilin acts as a stabiliser and anchors actin filaments at the Z-disc, through a myotilin/actin/α-actinin ternary complex, and thus is fundamentally important for the myofibrillar organisation of the Z-disc and may have roles in sarcomerogenesis [124]. Myotilin also interacts with the MuRF ubiquitin ligases and signalling cascade regulators [125]. Despite these crucial functions, a global inactivation of the *Myot* gene in mice showed no overt phenotype [126].

#### 2.4.1. Myotilin in Myopathy

*MYOT* variants mainly manifest in myopathies such as LGMD, MFM, distal myopathy and spheroid-body myopathy [127,128,129,130,131] (Table 4). Such diseases caused by pathogenic variants in *MYOT* are present in 10% of MFM patients and have been termed “myotilinopathies” [132]. Myotilinopathy variants usually result in structural changes of the Z-disc and the formation of polymorphous aggregates [122,127,133,134].

Almost all *MYOT* variants have been located in exon 2 in the amino-terminal region, an intrinsically unstructured region [135]. Two separate studies described *MYOT* missense variants affecting the same residue (p.Arg6Gly and p.Arg6His), both causing severe forms of skeletal myopathy, suggesting this residue is a mutational hotspot [136,137]. Further, patients carrying *MYOT* p.Arg6Gly as homozygous variant had a more severe phenotype, suggesting zygosity differences could play a role [137]. Similar to the *MYOT* p.Arg6Gly variant, despite being a dominant-negative variant, a more severe phenotype was reported in patients carrying the homozygous *MYOT* p.Ser60Phe mutation [138]. The only variant discovered outside of exon 2 is *MYOT* p.Arg405Leu, which is found in exon 9 and affects the second Ig domain and causes LGMD1A [130]. It was suggested that it inhibits myotilin homodimerization thus results in reduced interaction with α-actinin [130].

Z-disc streaming is reported in myotilinopathies along with the accumulation of dense myotilin and desmin-containing protein aggregates. These aggregates are likely caused by a reduction in protein turnover and myotilin degradation via the ubiquitin proteasome system [139]. However, due to the variants being located away from the binding site regions for α-actinin and filamin C, Kostan et al. found no evidence of altered actin binding—a causative factor for protein aggregation. Instead, these missense variants may affect the dynamics of Z-disc protein interactions and cause disruption to the maintenance and structural integrity of the Z-discs. Therefore, it was proposed that aggregation may be promoted by pathogenic myotilin variants exposing a hydrophobic cluster since most variants are substitutions from a polar to a hydrophobic residue [122,139,140].

Proteomics analysis on patients with *MYOT* p.Leu36Glu, p.Ser60Cys and p.Ser60Phe found that the Z-disc proteins myotilin, desmin, filamin C and other interacting partners (BAG3, HSPB8, p62) accounted for more than 70% of over-represented proteins in the aggregates indicating that myotilin-related MFM aggregates are structured formations from the Z-disc [141]. Interestingly, filamin C and desmin were more abundant than myotilin.

LGMDA1 patients exhibit disorganised sarcomere striations and Z-disc streaming [122]. This effect is also observed in disease caused by pathogenic variants in other Z-disc proteins, such as telethonin.

Since myotilin aggregation is common in myotilinopathy (like other MFMs), mouse models have been used to better understand the pathological effect. In models overexpressing wildtype myotilin, myofibrillar aggregation was reported along with severe muscle degeneration as found in LGMD1A [126,142]. However, investigating myotilin variants in transgenic mice has proved difficult as many cannot reproduce an in vivo phenotype [143]. Furthermore, zebrafish and murine in vivo electroporation methods have been utilised to characterise the function of myotilin variants [143,144].

#### 2.4.2. Summary

Although the exact pathomechanisms behind myotilinopathies are yet to be completely elucidated, missense variants in *MYOT*, which typically cause MFM via an accumulation of protein aggregates may be due to an impaired protein degradation system linked to Z-disc structural abnormalities [145]. Myotilinopathies have a clear role in myopathy, however patients often present additionally with a cardiac phenotype. For instance, the *MYOT* p.Ser60Phe variant has been described in a case of myopathy with low ejection fraction and cardiomyopathy, while other patients carrying the variant displayed with cardiomyopathy, left bundle branch block and congestive heart failure [127]. Despite this, no isolated cardiomyopathy cases have been attributed to variants in *MYOT*.

**Table 4 ijms-22-03058-t004:** Variants in the *MYOT* gene that have been previously reported in individuals with myopathy or cardiomyopathy.

Name Gene	Variant	Het or Homo	Type of Disease	Cardiac or Muscular	* MAF on GnomAD	Location	Ref
Myotilin *MYOT*	**p.Arg6His**	**Het**	**LGMD1A**	**Muscular**	**2.63 × 10^−5^**	**Serine rich N-terminus**	[136]
	**p.Arg6Gly**	**Homo**	**Severe MFM**	**Muscular**	**Absent**	**Serine rich N-terminus**	[137]
	**p.Ser39Phe**	**Het**	**Spheroid body myopathy**	**Muscular**	**Absent**	**Serine rich N-terminus**	[133]
	**p.Ser55Phe**	**Het**	**LGMD1A**	**Muscular**	**Absent**	**Serine rich N-terminus**	[120]
	**p.Thr57Ile**	**Het**	**LGMD1A**	**Muscular**	**Absent**	**Serine rich N-terminus**	[122]
	**p.Ser60Cys**	**Het**	**LGMD1A**	**Muscular**	**Absent**	**Serine rich N-terminus**	[127]
	**p.Ser60Phe**	**Het**	**LGMD1A**	**Muscular**	**1.31 × 10^−5^**	**Serine rich N-terminus**	[127]
	**p.Ser95Ile**	**Het**	**LGMD1A**	**Muscular**	**Absent**	**Serine rich N-terminus**	[127]
	**p.Arg405Lys**	**Het**	**LGMD1A**	**Muscular**	**Absent**	**Ig2**	[130]

Abbreviations: Het, heterozygous; Homo, homozygous; LGMD, limb-girdle muscular dystrophies; MFM, myofibrillar myopathy; N-terminus, amino-terminus. * MAF on GnomAD: The Minor Allele Frequency (MAF) was found on The Genome Aggregation Database (gnomAD) version 3.1. All variants with MAF < 10^−4^ are highlighted in bold.

### 2.5. Telethonin

Telethonin (also called TCAP) is found in both human heart and skeletal muscle and is especially abundant in skeletal muscle [146]. It is one of the Z-disc proteins that help to form the stable sarcomere structure and is also required for sarcomerogenesis [147].

Telethonin has been found to interact with titin and FATZ in the sarcomere. The first 140 amino acids of telethonin interact with the Ig domains Z1 and Z2 in the amino-terminal repeat region of titin [148] (Figure 5), leading to a tertiary complex of telethonin with titin and muscle LIM protein; this complex was suggested as a mechanical stretch sensor [149]. Additionally, telethonin interacts with calsarcin-1, which has a role in tethering the phosphatase calcineurin to the Z-disc [150]. Telethonin is encoded by the gene *TCAP*, which is located on chromosome 17q12.

*Tcap* knock-out mice do not display a cardiac phenotype under normal conditions [149]. However, upon biomechanical stress, telethonin deficiency leads to enhanced p53 levels and promotes apoptosis and cell death, suggesting initiation of heart failure development [149].

#### 2.5.1. Telethonin in Cardiomyopathy

Hayashi et al. analysed *TCAP* in 346 patients with HCM and 136 patients with DCM and identified the *TCAP* variant p.Thr137Ile in a HCM patient, and *TCAP* p.Glu132Gln in a patient with DCM [150]. Binding assays suggested that the HCM-associated variant reduced the telethonin’s binding to Z-disc proteins calsarcin-1 and titin, while the DCM variant additionally impacted binding to muscle LIM protein. However, also *TCAP* p.Arg153His showed alterations in binding properties, despite the fact that it is relatively common to be pathogenic (Table 5).

Two further *TCAP* variants (p.Arg70Trp and p.Pro90Leu) were discovered in a cohort of 389 HCM patients [151]. They also identified *TCAP* p.Glu13del, which is now assumed not to be pathogenic due to its frequency in normal cohorts [152,153].

Moreover, a novel *TCAP* p.Cys57Trp variant was identified in a small family who presented with HCM [154]. The variant co-segregated with disease in affected family members and 11 common HCM disease genes were excluded. The variant was found to have a very low population frequency and was highly conserved across different species. Seven bioinformatic tools suggested that the variant was likely to be pathogenic and therefore damaging [154]. The variant was located in the muscle LIM Protein interacting domain.

Bidirectional sequencing of six genes from samples of 313 patients with idiopathic DCM, revealed three *TCAP* variants from 3 different probands [155]. Two of these variants were found to have a possible disease association (*TCAP* p.Arg18Gln and p.Glu49Lys).

Hirtle-Lewis et al. interrogated *TCAP* by Sanger sequencing to analyse DNA from 96 patients with DCM and 184 control samples [156]. *TCAP* p.Arg158Cys was thought to be of clinical significance and due to the damaging nature of the change.

13 unrelated patients, who underwent heart transplantation due to end-stage DCM, were screened for genetic variants in 15 candidates genes using PCR and next-generation sequencing (NGS). *TCAP* p.Glu105Gln was identified in the cohort, but its frequency in normal cohorts makes a pathogenic role doubtful [157].

In summary, a small number of missense variants in *TCAP* have been associated with either DCM or HCM, but not all variants initially identified would withstand more recent scrutiny based on their frequencies in normal cohort (e.g., *TCAP* p. Glu13del). Altered binding properties to interactors such as muscle LIM Protein, titin and calsarcin-1 have been suggested as modes of action, but all conclusions so far are based on in vitro experiments only.

#### 2.5.2. Telethonin in Myopathy

Two *TCAP* variants were identified in patients from three families who presented with LGMD 2G. Their symptoms were variable and included distal muscle weakness in the legs, difficulty walking and severe calf hypertrophy [158]. One truncating variant *TCAP* p.Gln53X was found to be homozygous in two of the families, but heterozygous in another. A second variant was found to segregate with the third affected family and led to the deletion of two guanine nucleotides (in the genomic sequence these were nt 637–640). This variant was predicted to lead to a frameshift mutation resulting in a premature stop codon (*TCAP* p.Gly37fsX).

The homozygous variant *TCAP* p.Trp25X.IA was identified in a 27-year-old woman who presented with LGMD 2G [159]. Analysis of a muscle biopsy using immunohistochemistry (IHC) and Western blotting, showed a complete lack of telethonin in the skeletal muscle in the patient.

A cohort study of 300 patients with autosomal recessive LGMD identified eight patients with LGMD 2G, who either had no or reduced telethonin protein levels [160]. Clinical features were predominantly progressive proximo-distal muscle weakness, scapular winging and calf hypertrophy. Genetic analysis in the eight patients revealed a novel *TCAP* variant *TCAP* p.Ser11X and the previously reported one *TCAP* p.Arg12fs31X [161].

Barresi et al. described a male 49-year-old who presented with progressive LGMD. IHC using an antibody directed at the aa 58–167 (carboxy-terminal end) of telethonin, revealed no signal, and sequencing identified the homozygous *TCAP* p.Gln82X variant [162]. However, when an antibody against the full-length protein was used, the truncated telethonin protein was detected by Western blotting, although the signal appeared weaker than for the control. A muscle biopsy from the patient revealed co-localisation of the truncated protein with filamin C at the Z-discs, suggesting the truncated telethonin variant can still be incorporated into the sarcomere.

Another case study described a 35-year-old female Turkish patient with LGMD 2G [163]. A muscle biopsy revealed myopathic muscle with atrophy of the fibres and complete absence of telethonin in Western blotting. Using NGS, the novel homozygous frame-shift variant in *TCAP* (p.Ser31HisfsX11) was identified, resulting in truncation and/or nonsense mediated decay of the telethonin protein [163]. The same variant was detected by NGS in a 37-year-old Greek female LGMD 2G patient [164]. Western blot analysis revealed a complete lack of telethonin. Single nucleotide polymorphism analysis of a 10 Mb genomic region containing the *TCAP* gene, found a shared homozygous haplotype of the Greek and Turkish patients [164]. This suggests the possibility of a founder effect for that region of the Mediterranean for the *TCAP* variant described.

18 individuals from a minority of Bulgarian Muslims were found to be homozygous for *TCAP* p.Trp25X [165]. The authors looked at 100 new-borns in the area and found the heterozygous carrier rate of p.Trp25X to be 2%, meaning the expected homozygous patient number would be 1 in 2500 births in the region. Clinical symptoms included weakness in proximal muscle in the legs, supported by an MRI, which also progressed to the upper limbs. Similar to the case study presented by [159], Western blot analysis showed complete absence of the telethonin protein.

Yee et al. [166] identified one novel splice variant (*TCAP* c.110 + 5G > A) and one previously reported variant (*TCAP* p.Glu12ArgfsX20) [167] in a Chinese cohort. The proband of one family was homozygous for the splice variant c.110 + 5G > A, whilst three members of a second family were heterozygous for both variants. The proband from a third family was found to be homozygous for *TCAP* p.Glu12ArgfsX20. Clinical features, muscle MRI and muscle histopathology were used to support the diagnosis and indicate a genotype-phenotype relationship.

#### 2.5.3. Summary

Several studies report a link between variations in *TCAP* and HCM and DCM. Some of these variants affect conserved residues and either have a low allele frequency or do not appear in GnomAD. They are therefore considered as (potentially) pathogenic. Further co-segregation and functional studies are needed to support these findings and understand the mechanisms of pathogenicity. However, many of the initially reported variants have an allele frequency higher than 10^−4^ and hence appear relatively frequently in the general population for disease causing variants (Table 5).

A number of studies indicate a causative role of homozygous *TCAP* variants resulting in premature stop codons in LGMD 2G. Analyses of skeletal muscle biopsies suggest lack of functional telethonin protein [159,160,161,162,163,164,165,166,167,168]. However, findings depend on the epitopes of the antibodies used, so it is currently not clear how much nonsense mediate decay contributes to the disease mechanism. The variants described have low allele frequencies or are absent in GnomAD, suggesting they do contribute to pathogenicity (Table 5). Mechanistic studies on *TCAP* knockout mice support a role of *TCAP* in LGMD [168].

**Table 5 ijms-22-03058-t005:** Variants in the *TCAP* gene that have been previously reported in individuals with myopathy or cardiomyopathy.

Name	Variant	Het or Homo	Type of Disease	Cardiac or Muscular	* MAF on GnomAD	Location	Ref
Telethonin *TCAP*	**p.Ser11X**	**Homo**	**LGMD2G**	**Muscular**	**6.57 × 10^−6^**		[160]
	**p.Arg12fs31X**	**Homo**	**LGMD2G**	**Muscular**	**Absent**		[161]
	**p.Glu12ArgfsX20**	**Homo**	**LGMD2G**	**Muscular**	**1.31 × 10^−5^**		[166]
	p.Glu13del	Het	HCM	Cardiac	9.99 × 10^−4^	Titin-binding	[151]
	**p.Arg18Gln**	**Het**	**DCM**	**Cardiac**	**2.39 × 10^−5^**	**Titin-binding**	[155]
	**p.Trp25X**	**Homo**	**LGMD2G**	**Muscular**	**6.57 × 10^−6^**		[159]
	**p.Ser31HisfsX11**	**Homo**	**LGMD2G**	**Muscular**	**Absent**		[163]
	**p.Gly37fsX**	**Het**	**LGMD2G**	**Muscular**	**Absent**		[158]
	**p.Glu49LyS**	**Het**	**DCM**	**Cardiac**	**3.70 × 10^−5^**	**Titin-binding**	[155]
	**p.Gln53X**	**Homo/Het**	**LGMD2G**	**Muscular**	**1.31 × 10^−5^**		[158]
	**p.Cys57Trp**	**Het**	**HCM**	**Cardiac**	**2.20 × 10^−5^**	**Titin-binding**	[154]
	**p.Arg70Trp**	**Het**	**HCM**	**Cardiac**	**4.62 × 10^−5^**	**Titin-binding**	[151]
	**p.Gln82X**	**Homo**	**LGMD2G**	**Muscular**	**Absent**		[162]
	**p.Pro90Leu**	**Het**	**HCM**	**Cardiac**	**4.10 × 10^−6^**	**linker**	[151]
	p.Glu105Gln	Het	DCM	Cardiac	5.10 × 10^−4^	linker	[157]
	**p.Glu132Gln**	**Het**	**DCM**	**Cardiac**	**4.01 × 10^−6^**	**C-terminus**	[150]
	**p.Thr137Ile**	**Het**	**HCM**	**Cardiac**	**Absent**	**C-terminus**	[150]
	p.Arg153His	Het	HCM	Cardiac	2.37 × 10^−4^	C-terminus	[150]
	**p.Arg158Cys**	**Het**	**DCM**	**Cardiac**	**Absent**	**C-terminus**	[156]
	**c.110 + 5G > A**	**Homo**	**LGMDR7**	**Muscular**	**Absent**		[167]

Abbreviations: DCM, dilated cardiomyopathy; HCM, hypertrophic cardiomyopathy; Het, heterozygous; Homo, homozygous; LGMD, limb-girdle muscular dystrophies. * MAF on GnomAD: The Minor Allele Frequency (MAF) was found on The Genome Aggregation Database (gnomAD) version 3.1. All variants with MAF < 10^−4^ are highlighted in bold.

### 2.6. ZASP/Cypher

ZASP is a member of the enigma family, which are structural Z-disc proteins with signalling potential. The murine homologue of ZASP is called Cypher, when discussing literature pertaining to both human and murine isoforms, the name ZASP will be used. ZASP is a cytoskeletal protein, which is highly important in maintaining Z-disc structural and muscular integrity by forming multiprotein complexes binding to α-actinin and could be involved in signalling roles via interactions with protein kinase C [169].

ZASP is encoded by the *LDB3* gene on chromosome 10q22.3-10q23.2 and consists of 16 exons encoding six distinct isoforms, which are highly expressed in human striated muscles [170]. There is evidence of alternative splicing: ZASP1, ZASP5 and ZASP8 possess exon 4 and are predominantly expressed in cardiac tissue, whereas the ZASP isoforms that have exon 6 (ZASP2 and ZASP6) are more highly expressed in skeletal muscle [171].

Being a member of the enigma family, ZASP is composed of an amino-terminal PDZ motif and between 0 and 3 LIM domains at the carboxy-terminus (Figure 6), hence the alternative name: LIM domain-binding protein 3 (*LDB3*). Both PDZ and LIM domains are involved in protein-protein interactions [172,173]. Along with other PDZ-LIM domain proteins, actinin-associated LIM protein and carboxy-terminal LIM protein, ZASP has a conserved region named the ZASP-like motif (ZM), found between the PDZ and LIM domains. This motif is present in both cardiac-specific and skeletal muscle predominant ZASP isoforms, mediates binding to actin and can mediate self-interaction of ZASP proteins [174,175,176]. ZASP interacts with protein kinase C, α-actinin, myotilin, telethonin and FATZ/calsarcin/myozenin in striated muscle and plays roles in myofibrillogenesis and assembling protein complexes and integrin adhesion sites within the muscle Z-disc [124]. ZASP also binds to Ankrd2 and p53 in a trimeric complex, which may be involved in signalling, regulation of gene expression and muscle differentiation [171].

Ablation of Cypher in a mouse model resulted in neonatal lethality after birth exhibiting severe congenital myopathy phenotypes and DCM with heart failure [169]. Further to this, in Cypher knockout mice, Z-disc abnormalities such as disorganisation and fragmentation were present in both skeletal and cardiac muscle after birth inferring that Cypher is an important player in the maintenance of the Z-disc during contraction [169]. The cardiac phenotype was later supported by a zebrafish morphilino knockdown of Cypher, which resulted in DCM phenotypes [177].

#### 2.6.1. ZASP in Cardiomyopathy

*LDB3* variants were first described in patients with familial and sporadic DCM, LVNC, HCM and ACM [18,178,179,180]. Due to the alternative splicing of ZASP and their resulting tissue-specific isoform expression, the exon location of each variant tends to specify the myopathy type (cardiac/skeletal) [172]. For instance, patients with variants found in the cardiac-specific exon 4 (*LDB3* p.Thr213Ile and p.Ser196Leu) present with DCM and LVNC. This exon encodes the LIM domain of ZASP, therefore variants may affect cardiac function by altering the binding affinity of ZASP to its binding partners [18,181]. However, a biophysical characterisation of *LDB3* p.Ala157Thr, p.Ala165Thr and p.Arg268Cys found that these ZM-domain variants did not affect the affinity the of ZASP to G-actin [182].

HCM patients have also been found to carry *LDB3* variants, which localise between exon 10 and 13 (*LDB3* p.Ser196Leu, p.Asp366Asn, p.Tyr468Ser and p.Pro615Leu) [18,178]. DCM variants in exon 4 and 10 showed impaired binding to phosphoglucomutase, yet variants in exon 6, which is not limited to the heart, had no effect on this interaction [183].

Further, it was documented that the DCM-associated *LDB3* p.Asp117Asn variant affects interactions within the actin cytoskeletal network in vitro, while the *LDB3* p.Asp626Asn variant increases its binding affinity to protein kinase C [179]. While there are suggestions that ZASP variants may result in observed patient phenotypes via altered protein kinase C binding affinity, the underlying molecular mechanisms are yet to be elucidated.

#### 2.6.2. ZASP in Myopathy

As mentioned previously, animal models found that ablation of Cypher results in a severe skeletal muscle defects [169]. This is also true in humans. Pathogenic variants of the human *LDB3* gene are reported in skeletal myopathies such body myositis [18,178,184]. Carriers of skeletal muscle-affecting *LDB3* variants usually present with muscle weakness more distally than proximally [185]. The first identified *LDB3* variants with direct skeletal muscle involvement were heterozygous missense variants (Exon 6: *LDB3* p.Ala147Thr, p.Ala165Val. Exon 9: *LDB3* p.Arg268Cys) found in 11 MFM patients. Since then, less than 50 patients have been identified with ZASP-MFM [132,184,186,187,188,189,190]. Like in cardiomyopathy, the loci may determine the disease outcome: skeletal muscle myopathies dominate when the variant is located in exon 6 (MFM: *LDB3* p.Ala165Val and p.Ala147Thr) though a cardiac phenotype has also been reported (DCM/LVNC: *LDB3* p.Asp117Asn and p.Lys136Met) [178,181,186].

Out of these *LDB3* p.Ala147Thr and p.Ala165Val are by far the most common variants found in ZASP-MFM. This variant hotspot (also containing the *LDB3* p.Asn155His variant [189]) is located in the actin-binding ZM domain. In the presence of these variants, the actin-binding domain is dysfunctional and causes actin disruption along the Z-disc, affecting Z-disc and myofibril integrity [176,190].

These patients displayed sarcolemmal invaginations and aggregates containing ZASP, ubiquitin, p62 and LC3 as well as the accumulation of autophagic vacuoles, inferring that aggregation and autophagy are potentially additional features of zaspopathy [190].

Human myotonic dystrophy patients often present with mis-spliced ZASP isoforms in their skeletal muscle [191]. A Drosophila model of this disease found splicing abnormalities in the Drosophila orthologue of Cypher along with Z-disc abnormalities suggesting that irregular Cypher splicing may be a cause of human myotonic dystrophy [192].

#### 2.6.3. Summary

Variants of *LDB3* have been associated with both cardiomyopathies and MFM. In particular, autosomal dominant missense variants of the *LDB3* gene are linked with ZASP-related MFM [193]. These variants have been reported to disrupt the Z-disc and result in increased myofibrillar aggregation. Variants were found to primarily affect ZASP in the PDZ and LIM domains.

## 3. Conclusions and Future Directions

As demonstrated in this review, the Z-disc proteins α-actinin, filamin C, myopalladin, myotilin, telethonin and ZASP do not only play important roles in anchoring actin-filaments and relaying signals at the Z-disc signalling hub, but they are also required for the integrity of skeletal muscle and the heart. Pathogenic variants in all six proteins can cause myopathies and/or cardiomyopathies (summarised in Table 1, Table 2, Table 3, Table 4, Table 5 and Table 6).

In the last decade, the use of high-throughput sequencing techniques has revolutionised clinical genetics in several ways: geneticists now have a better understanding of genomic variation in normal cohorts [194], which helps to evaluate the pathogenicity of variants identified in patients based on their frequency. For cardiomyopathy variants, which are rare genetic diseases, a threshold of MAF < 10^−4^ is now consensus [19]. Based on this, a critical review of many, especially missense variants, which were identified before the application of high-throughput sequencing, would now consider them relatively frequent to be pathogenic (see Table 1, Table 2, Table 3, Table 4, Table 5 and Table 6). Moreover, high-throughput sequencing allows the cost-efficient interrogation of gene panels or whole exomes and may help to identify potentially pathogenic variants in patients. However, an overwhelming challenge in the clinical practice is the evaluation of these variants, especially of missense ones. Co-segregation studies in families are valuable and bioinformatics tools can provide additional support.

For several genes, especially *FLNC*, it is not clear why some truncating or missense variants cause pure myopathy while others cause cardiomyopathy without skeletal muscle involvement. It is tempting to speculate that there are functional differences between the variants. Future functional studies will hopefully be able to shed light on this and application of exciting novel technologies will facilitate such studies: induced pluripotent stem cells allow the generation of human cardiac and skeletal muscle cells from patients [195]. Moreover, genome-engineering allows the precise manipulation of genomes, both in animal models and induced pluripotent stem cells [196]. Hence, researchers will be able to generate accurate models of human disease and interrogate them for open questions in the field. This will help to develop a deeper understanding of pathogenic mechanisms and thereby facilitate the development of novel therapeutic approaches to benefit the patients.

## Figures and Tables

**Figure 1 ijms-22-03058-f001:**
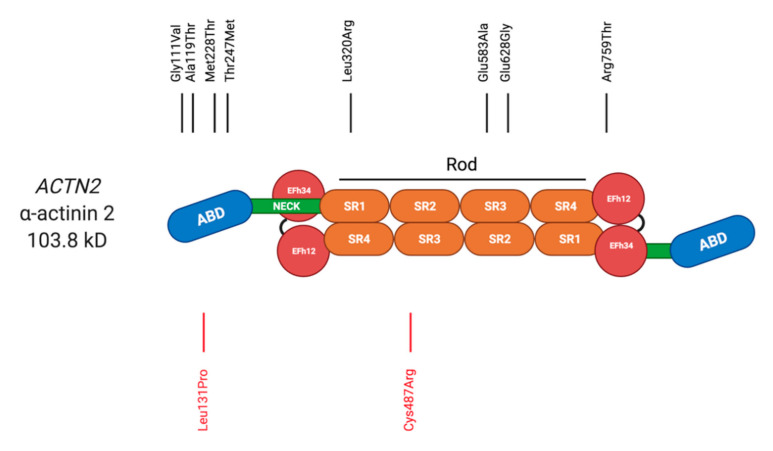
Schematic representation of α-actinin shown as a dimer with variants in the *ACTN2* gene that have been previously reported in individuals with myopathy (red) or cardiomyopathy (black). Solid lines represent missense variants. ABD—actin binding domain, EFh—EF hand, SR—spectrin like repeat. Created with BioRender.com (accessed on 7 March 22021). Adapted from [16].

**Figure 2 ijms-22-03058-f002:**
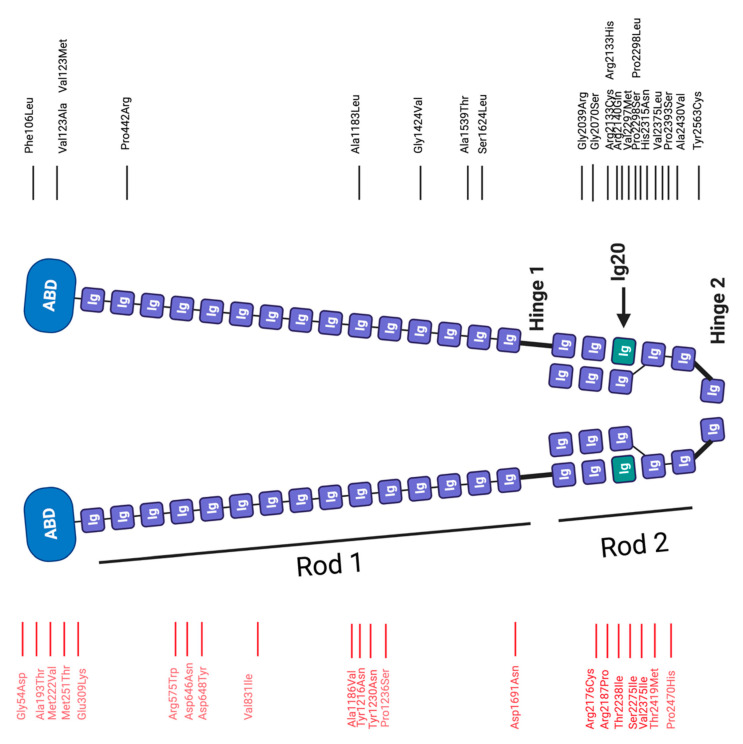
Schematic representation of Filamin C shown as a dimer with missense variants in the *FLNC* gene that have been previously reported in individuals with myopathy (red) or cardiomyopathy (black). ABD—actin binding domain, Ig—immuno-globulin like domain. For truncating variants, please refer to Table 2. Created with BioRender.com (accessed on 7 March 2021).

**Figure 3 ijms-22-03058-f003:**
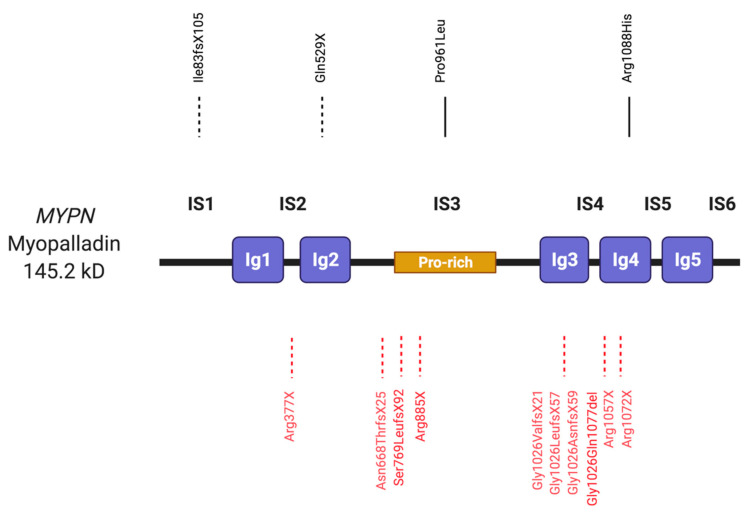
Schematic representation of Myopalladin shown as a monomer with variants in the *MYPN* gene that have been previously reported in individuals with myopathy (red) or cardiomyopathy (black). Solid lines represent missense variants and dashed lines are truncations. Ig—immuno-globulin like domain, IS—interdomain insertion, Pro-rich—Proline-rich. Created with BioRender.com (accessed on 7 March 2021).

**Figure 4 ijms-22-03058-f004:**
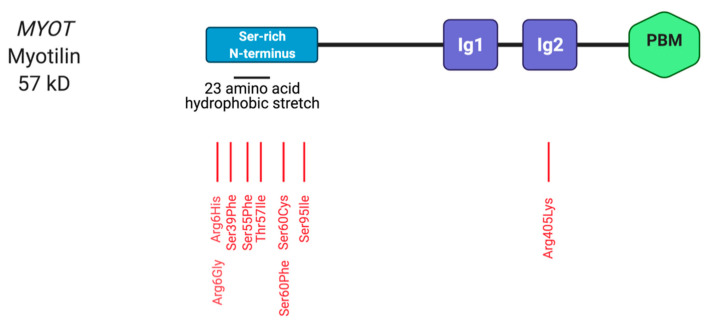
Schematic representation of Myotilin shown as a monomer with variants in the *MYOT* gene that have been previously reported in individuals with myopathy (red). Solid lines represent missense variants. Ig—immuno-globulin like domain, Ser-rich—Serine-rich, PBM—PDZ-binding motif. Created with BioRender.com (accessed on 7 March 2021).

**Figure 5 ijms-22-03058-f005:**
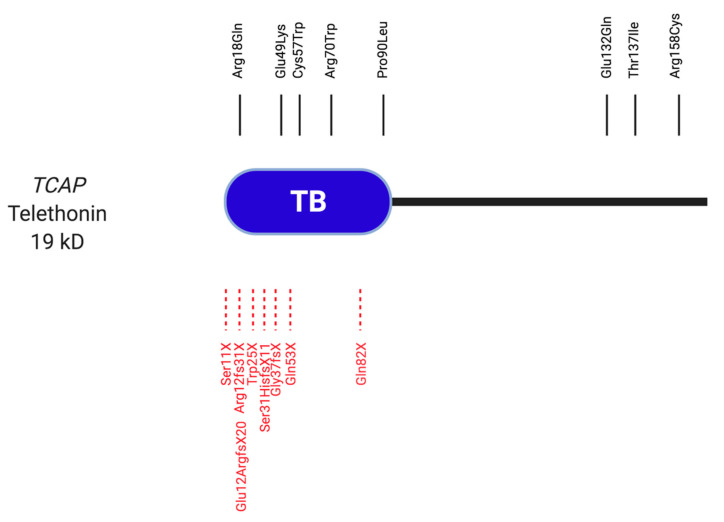
Schematic representation of Telethonin shown as a monomer with variants in the *TCAP* gene that have been previously reported in individuals with myopathy (red) or cardiomyopathy (black). Solid lines represent missense variants and dashed lines are truncations. TB—titin binding domain. Created with BioRender.com (accessed on 7 March 2021).

**Figure 6 ijms-22-03058-f006:**
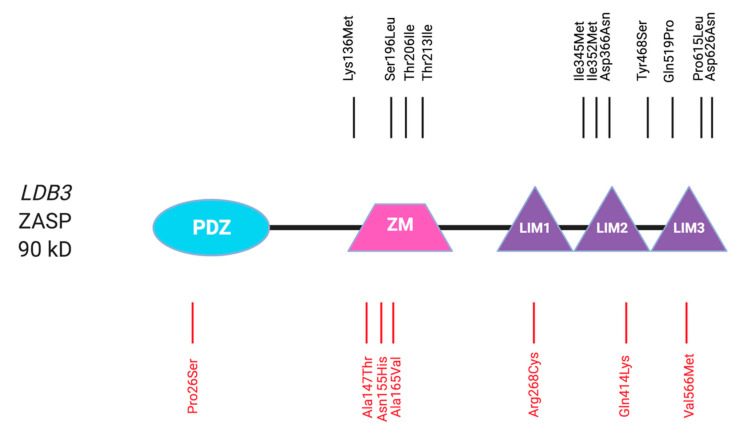
Schematic representation of ZASP shown as a monomer with variants in the LDB3 gene that have been previously reported in individuals with myopathy (red) or cardiomyopathy (black). Solid lines represent missense variants. LIM—Lin-11 Isl-1 Mec-3 (LIM) domain, PDZ—PDZ domain, ZM—ZASP-like motif. Created with BioRender.com (accessed on 7 March 2021).

**Table 6 ijms-22-03058-t006:** Variants in the *LDB3* gene that have been previously reported in individuals with myopathy or cardiomyopathy.

Name Gene	Variant	Het or Homo	Type of Disease	Cardiac or Muscular	* MAF on GnomAD	Location	Ref
ZASP *LDB3*	**p.Pro26Ser**	**Het**	**MFM**	**Muscular**	**Absent**	**PDZ motif**	[190]
	p.Asp117Asn	Het	DCM, LVNC	Cardiac	7.10 × 10^−3^	ZM motif	[178]
	**p.Lys136Met**	**Het**	**DCM, LVNC**	**Cardiac**	**Absent**	**ZM motif**	[178]
	**p.Ala147Thr**	**Het**	**MFM**	**Muscular**	**Absent**	**ZM motif**	[186]
	**p.Asn155His**	**Het**	**Distal MFM**	**Muscular**	**Absent**	**ZM motif**	[186]
	**p.Ala165Val**	**Het**	**MFM, DCM, Markesbery**	**Both**	**Absent**	**ZM motif**	[189]
	p.Ala171Thr	Het	DCM	Cardiac	1.51 × 10^−4^	ZM motif	[171]
	p.Ser189Leu	Het	DCM	Cardiac	5.19 × 10^−4^	ZM motif	[178]
	**p.Ser196Leu**	**Het**	**DCM, LVNC, HCM**	**Cardiac**	**6.57 × 10^−6^**	**ZM motif**	[178]
	**p.Thr206Ile**	**Het**	**DCM**	**Cardiac**	**6.57 × 10^−6^**	**ZM motif**	[178]
	**p.Thr213Ile**	**Het**	**DCM, LVNC**	**Cardiac**	**Absent**	**ZM motif**	[178]
	p.Ala222Thr	Het	MFM	Muscular	3.48 × 10^−4^	ZM motif	[88]
	**p.Arg268Cys**	**Het**	**MFM**	**Muscular**	**4.61 × 10^−5^**	**LIM domain**	[186]
	**p.Ile345Met**	**Het**	**DCM**	**Cardiac**	**Absent**	**LIM domain**	[178]
	**p.Ile352Met**	**Het**	**DCM**	**Cardiac**	**Absent**	**LIM domain**	[178]
	**p.Asp366Asn**	**Het**	**HCM**	**Cardiac**	**Absent**	**LIM domain**	[18]
	**p.Gln414Lys**	**Het**	**MFM**	**Muscular**	**4.60 × 10^−5^**	**LIM domain**	[88]
	**p.Tyr468Ser**	**Het**	**HCM**	**Cardiac**	**Absent**	**LIM domain**	[18]
	**p.Gln519Pro**	**Het**	**HCM**	**Cardiac**	**Absent**	**LIM domain**	[18]
	**p.Val566Met**	**Het**	**Distal dominant weakness**	**Muscular**	**2.63 × 10^−5^**	**LIM domain**	[184]
	**p.Pro615Leu**	**Het**	**HCM**	**Cardiac**	**Absent**	**LIM domain**	[18]
	**p.Asp626Asn**	**Het**	**DCM**	**Cardiac**	**Absent**	**LIM domain**	[179]

Abbreviations: DCM, dilated cardiomyopathy; HCM, hypertrophic cardiomyopathy; Het, heterozygous; Homo, homozygous; LVNC, left ventricular noncompaction; MFM, myofibrillar myopathy. * MAF on GnomAD: The Minor Allele Frequency (MAF) was found on The Genome Aggregation Database (gnomAD) version 3.1. All variants with MAF < 10^−4^ are highlighted in bold.

## Data Availability

Data sharing not applicable.

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
