# Peer review of "The Role of Z-disc Proteins in Myopathy and Cardiomyopathy"

_ijms, 2021, doi:10.3390/ijms22063058_

Round 1

Reviewer 1 Report

The authors reviewed the role of Z-disc proteins in skeletal muscle and cardiac disease. The review is well written and easy to read, and is very informative, clear and concise.

I have only some minor comments and suggestions as follows

Line 24: remove legs as it is lower limb and comes under distal myopathy and not proximal as mentioned in the parentheses.

Lines 24 and 25: Is LGMDA an example of proximal muscle disease  - need to rephrase the sentence.

Line 32: (NM), which...

Line 38: cardiomyopathies and left ventricular...

Line 39: (systolic dysfunction), while HCM...

Lines 37 to 41: is a complex sentence, and would be great to make it into simple sentences.

Line 45: a single sentence is written as a paragraph. Kindly merge it with the previous paragraph.

Line 47: The striated muscle cells,....

Lines 65-68: Reference is missing

Line 83: instead of is too common, it would be better to describe it as "to be relatively common"....

Line 84: HCM-associated...

Lines 91-97: merge paragraphs

Line 93: an another variant...

Line 106: to muscle LIM protein...

Line 110: co-segregated 

Table 1: Expand Het or Homo or include it in the abbreviations used below the table.

Line 117: Abbreviations appear twice, delete one of them

Line 120: OM, other non-specified myopathy

Line 129: Tibialis anterior muscle,...

Line 134: maintain knockout for consistency

Line 145: 2.2 Filamin C in bold letters

Line 159: [36], and is thought to....

Line 162: Ig24, which...

Line 169: filamin C includes...

Line 171: can be modulated....

Line 190: introduce space between involvement and the references

Line 197: close the space between 95 and %

Line 199: introduce space between filaminopathies and the reference

Line 213: introduce space between Zz-disc and the references

Line 214: mouse C2C12 myoblasts..

Line 220-221: Reference missing

Line 223: introduce space between variants and the reference

Line 231: introduce space between characteristics and the reference

Line 237: exhibited defects in muscle structure and redistribution of .....

Line 243: Z-disc, which...

Line 245: chaperone-assisted...

Line 249: shown to cause...

Line 252: introduce space between Val and the reference

Line 257: introduce space between dementia and the references

Line 270: result in exclusive cardiac phenotype...

Line 271: Myopalladin in bold

Lines 275-278: reference missing

Line 283: [81, 83-86]

Line 296: cardiomyopathies. After screening...

Line 311: muscle LIM protein

Line 322: The fourth individual...

Line 338: loss-of-function

Line 341: full-length...

Line 353: Myotilin in bold

Line 354-355: reference missing

Line 358: cross-link actin

Line 361: stretch region, while...

Line 364: domains, which....

Line 365: introduce space between Ig domains and the reference

Lines 370 and 371: merge paragraphs

Line 375: Close one space between the period and Such diseases...

Line 385: a dominant-negative

Line 397: Therefore, it was...

Line 406: introduce space in between streaming and reference

Line 414: in vivo in italics for consistency with in vitro in italics elsewhere

Line 418: MYOT, which typically...

Line 423: cardiomyopathy, while...

Line 427: Telethonin in bold

Lines 429-430: reference missing

Line 434: muscle LIM protein

Line 438: knock-out

Lines 439-441: reference missing

Line 447: muscle LIM protein

Line 448: relatively common to be...

Line 452: [124,125]

Lines 4557-458: reference missing

Lines 458 and 472: muscle LIM protein

Line 496: an antibody...

Lines 503-505: reference missing

Line 507: Single nucleotide polymorphism

Lines 507-509: reference missing

Lines 534-535: reference missing

Line 540: ZASP/Cypher in bold

Line 544: protein, which is...

Line 548: isoforms, which are...

Line 561: striated muscle and plays...

Line 563: complex, which may be...

Line 571: Cypher, which...

Line 572: introduce space in between phenotypes and the reference

Lines 584-585: reference missing

Line 613: introduce space between zaspopathy and the reference

Lines 614-615: reference missing

Line 617: introduce space between dystrophy and the reference

Line 621: introduce space between MFM and the reference

Line 637: relatively frequent to be....

Author Response

We than the reviewer for their time and thorough consideration of our manuscript. We have addressed all comments, especially added the missing references.

Regarding Line 283: [81, 83-86]

This is a software glitch, which we unfortunately cannot rectify (Endnote keeps crashing). We have left a comment for the copy-editors to address.

Reviewer 2 Report

In this manuscript, “The role of Z-disc proteins in skeletal muscle and cardiac disease”, authors evaluated the effects of several Z-disc proteins on skeletal and cardiac muscles. There are several questions and suggestions I raised for this manuscript.

  1. In table 1, the part of “type of disease” is not easy to understand the separation of muscular and cardiac myopathy. Author can directly add another column to describe the disease is belong to muscular or cardiac myopathy. Another wire point in this table is the order of the variants in table. How the authors decide the order of these variants in one gene?

  1. Page 2 line 33, authors mentioned “Though generally classified as diseases of the skeletal muscle, patients with myopathies may additionally present with a cardiac phenotype (‘cardiac involvement’)”. Have any reference to prove this? Or authors can tell us why have this conclusion.

  1. Page 2 line 37, authors mentioned “Cardiomyopathies can be classified into dilated (DCM), hypertrophic (HCM), restrictive (RCM), arrhythmogenic (ACM) 38 cardiomyopathy and left ventricular noncompaction (LVNC), but there is phenotypic overlap between these diseases.”. However, authors did not say which phenotype is overlapped among these diseases.

  1. Please consistent the description of myopathy. For example, Line 57 (myopathies and cardiomyopathies) and Line 116 (muscular or cardiac myopathy).

  1. In table 1, some words are bold, but some are not. Have any meanings for this?

  1. 2.1 Diseases and 2.2.4 Summary could be combined to one section. 2.2.1 has no reason to be one section along.

  1. 4.2 Animal models has no reason to be one section along. This can be combined into 2.4.1, because this animal model is related to skeletal muscle diseases.

  1. Both line 442 and line 475 are labeled 2.5.1. Please check all of these.

  1. 6.1 Animal models can be combined into 2.6.2 or 2.6.3 to describe the animal models for skeletal or cardiac muscle diseases.

The overall organization needs to be improved.

Author Response

We thank the reviewer for their time and helpful suggestions.

In this manuscript, “The role of Z-disc proteins in skeletal muscle and cardiac disease”, authors evaluated the effects of several Z-disc proteins on skeletal and cardiac muscles. There are several questions and suggestions I raised for this manuscript.

In table 1, the part of “type of disease” is not easy to understand the separation of muscular and cardiac myopathy. Author can directly add another column to describe the disease is belong to muscular or cardiac myopathy. Another wire point in this table is the order of the variants in table. How the authors decide the order of these variants in one gene?

We have added a column as suggested. The variants are now shown in the order of amino acids. We have also taken up suggestion of reviewer 3 and separated the table per protein to make it more legible.

Page 2 line 33, authors mentioned “Though generally classified as diseases of the skeletal muscle, patients with myopathies may additionally present with a cardiac phenotype (‘cardiac involvement’)”. Have any reference to prove this? Or authors can tell us why have this conclusion.

 We have added a reference for the statement.

Page 2 line 37, authors mentioned “Cardiomyopathies can be classified into dilated (DCM), hypertrophic (HCM), restrictive (RCM), arrhythmogenic (ACM) 38 cardiomyopathy and left ventricular noncompaction (LVNC), but there is phenotypic overlap between these diseases.”. However, authors did not say which phenotype is overlapped among these diseases.

 There are many examples of phenotypic overlap between cardiomyopathies, e.g. between HCM and RCM (diastolic dysfunction) or between DCM and ACM (if the left ventricle is affected). To discuss this in detail is beyond the scope of this review. Nevertheless, we have expanded the sentence and added two references for the statement.

Please consistent the description of myopathy. For example, Line 57 (myopathies and cardiomyopathies) and Line 116 (muscular or cardiac myopathy).

 We now constituently used the terms myopathy for skeletal muscle disease and cardiomyopathy for heart muscle disease.

In table 1, some words are bold, but some are not. Have any meanings for this?

 We now state that all variants with MAF < 10-4 are in bold.

2.1 Diseases and 2.2.4 Summary could be combined to one section. 2.2.1 has no reason to be one section along.

 We agree with the reviewer and have revised the text accordingly.

4.2 Animal models has no reason to be one section along. This can be combined into 2.4.1, because this animal model is related to skeletal muscle diseases.

  We agree with the reviewer and have revised the text accordingly.

Both line 442 and line 475 are labeled 2.5.1. Please check all of these.

We have corrected this.

6.1 Animal models can be combined into 2.6.2 or 2.6.3 to describe the animal models for skeletal or cardiac muscle diseases.

  We agree with the reviewer and have moved the section.

The overall organization needs to be improved.

We have re-organised some of the text (see e.g. above) and put a figure and table for each protein, to make the structure easier to follow for the readers.

Reviewer 3 Report

This review has a detail description and is useful to the reader in this field. However, Table 1 is too long, and it is easy to read if it is dispersed in the description of each protein if possible. Moreover, it is helpful to understand that Fig. 1 can describe the mutation site.

Author Response

This review has a detail description and is useful to the reader in this field. However, Table 1 is too long, and it is easy to read if it is dispersed in the description of each protein if possible. Moreover, it is helpful to understand that Fig. 1 can describe the mutation site.

We thank the reviewer for their time, the positive evaluation and helpful suggestions. We have taken up the suggestion to split the table. We now present one table for each protein. Figure 1 has also been broken up into six figures, which all show the positions of the pathogenic variants (missense variants only for filamin C).

Round 2

Reviewer 2 Report

No more comments.